# Peer review of "Conidia Fusion: A Mechanism for Fungal Adaptation to Nutrient-Poor Habitats"

_jof, 2023, doi:10.3390/jof9070755_

Round 1
Reviewer 1 Report
The introduction or discussion section may have more about the nematode-trapping fungi mechanism and their relation to conidia fusion. Because the supplementary material includes trap structure, but there is not discussion around such relation: conidia fusion numbers, genera, etc.
The section 2.2.5 is not clear for me, how the stereomicroscope works to get those images, is coupled to a computer with a software named? company? how measurement are performed and density is measured compared to?
211. cfr of 3% es 10%
Reviewer 2 Report
Summary:
This study is on an interesting topic and a very interesting group of fungi. The introduction and discussion are thoughtfully written. The methods are straight-forward and the results are clearly communicated in text. However, it is clear Figure 6 has been manipulated to support the communicated result, as each panel—identified in the legend as different experimental groups—reflects the same branching pattern. This is highly concerning and draws into question the validity of the entire paper. Figure 4 additionally does not match written results, although this may be a simple labeling error.
Comments:
Ln 2: I suggest changing “the strategy” to “a strategy” in the title
Ln 100: The “full conidia productive period” is mentioned variously in the methods, but it is not fully defined. I understand this may vary between strains, but can you indicate a time-frame or observable characteristic that was used to identify this stage?
Figure 1: What does “Gap” signify in x-axis label? Should this be deleted?
Figure 3: I recommend adjusting the lettering such that B is between A and C.
Figure 4: The label of these figures are apparently reversed: the value for 0% in the text matches the value for 3% on the figure, and the trend communicated in text is reversed in the figure. Please correct this error.
Figure 6: This figure has been digitized, unlike Figure 5, and it’s clear that the path of mycelium in each panel match exactly. This pattern would be impossible from three separate treatments. Authors have further indicated different 6-day endpoints (in blue) along this identical mycelial network. This figure has been unethically manipulated, which draws into question not only the results of this section, but the entire paper.
Revisions:
Ln 18: remove double space after “strains”
Ln 76: change “Orbilimycetes” to “Orbiliomycetes”
Reviewer 3 Report
1. Improve the introduction focusing the effect of nutrient-poor habitats, using updated references;
2. Be sure to emphasized the treatment of the study per objective. What are those parameters needed to answer the objectives of the study;
3. Analyze an indicator that would really determine conidial fusion-physiologically and genetically- you should emphasized this.
4. Be consistent with writing styles in your methodology- lot of technical or mis-look on proper statement of each of the procedures and methods;
5. Improve presentation of results, check properly level of significance, letter designation labelling of x and y axis;
6. Recast the presentation of results with deeper understanding and presentation of justification which coincide with your results;
7. The paper can be improved and revision of the paper will be recommended

Be responsive on command of English grammar- using were and was
Round 2
Reviewer 2 Report
The author response for Figure 6 is not satisfactory. I recommend replacing this figure with photos, as in Figure 5, or a chart representing this data (e.g., a bar chart or box plot).
Reviewer 3 Report
Improved version of the paper
ok
